# High Temperature Alters Secondary Metabolites and Photosynthetic Efficiency in *Heracleum sosnowskyi*

**DOI:** 10.3390/ijms22094756

**Published:** 2021-04-30

**Authors:** Anna Rysiak, Sławomir Dresler, Agnieszka Hanaka, Barbara Hawrylak-Nowak, Maciej Strzemski, Jozef Kováčik, Ireneusz Sowa, Michał Latalski, Magdalena Wójciak

**Affiliations:** 1Department of Botany, Mycology and Ecology, Institute of Biological Sciences, Maria Curie-Skłodowska University, Akademicka 19, 20-033 Lublin, Poland; 2Department of Analytical Chemistry, Medical University of Lublin, Chodźki 4a, 20-093 Lublin, Poland; maciejstrzemski@umlub.pl (M.S.); i.sowa@umlub.pl (I.S.); magdalenawojciak@umlub.pl (M.W.); 3Department of Plant Physiology and Biophysics, Institute of Biological Sciences, Maria Curie-Skłodowska University, Akademicka 19, 20-033 Lublin, Poland; agnieszka.hanaka@poczta.umcs.lublin.pl; 4Department of Botany and Plant Physiology, University of Life Sciences in Lublin, Akademicka 15, 20-950 Lublin, Poland; barbara.nowak@up.lublin.pl; 5Department of Biology, University of Trnava, Priemyselná 4, 918 43 Trnava, Slovakia; jozkovacik@yahoo.com; 6Children’s Orthopedics Department, Medical University of Lublin, Gębali 6, 20-093 Lublin, Poland; michallatalski@umlub.pl

**Keywords:** heat stress, invasive plants, furanocoumarins, phenolic metabolism, photosynthesis

## Abstract

Due to global warming, invasive species have spread across the world. We therefore studied the impact of short-term (1 day or 2 days) and longer (7 days) heat stress on photosynthesis and secondary metabolites in *Heracleum sosnowskyi*, one of the important invasive species in the European Union. *H. sosnowskyi* leaves exposed to short-term heat stress (35 °C/1 d) showed a decrease in chlorophyll and maximum potential quantum efficiency of photosystem II (Fv/Fm) compared to control, 35 °C/2 d, or 30 °C/7 d treatments. In turn, the high level of lipid peroxidation and increased H_2_O_2_ accumulation indicated that the 30 °C/7 d stress induced oxidative damage. The contents of xanthotoxin and bergapten were elevated in the 2 d and 7 d treatments, while isopimpinellin was detected only in the heat-stressed plants. Additionally, the levels of free proline and anthocyanins significantly increased in response to high temperature, with a substantially higher increase in the 7 d (30 °C) treatment. The results indicate that the accumulation of proline, anthocyanins, and furanocoumarins, but not of phenolic acids or flavonols, contributes to protection of *H. sosnowskyi* plants against heat stress. Further studies could focus on the suppression of these metabolites to suppress the spread of this invasive species.

## 1. Introduction

Two of the greatest threats to biodiversity and ecosystem functioning are species invasions and global climate change [1]. Furthermore, climate change and invasions may interact, with climate-change conditions favoring and, thus, facilitating the spread of non-native species [2]. The mechanism underlying the interaction between climate change and biological invasions, however, remains unclear. One of the fundamental questions in invasion ecology is the issue of what makes an invading species successful. In order for invaders to become established in a recipient environment, they must first pass through the “ecological filter” of the environment [3]. Temperature is often regarded as the most important abiotic factor in determining species distribution owing to its impact on biochemical and cellular processes, which, in turn, affect organismal performance [4,5].

In particular, given the climate warming and prediction of global warming of 1.5 to 2.0 °C by 2050 [6], heat stress is currently considered one of the important factors that directly influence plant growth and development [7]. Exposure of plants to heat affects many aspects of vegetative processes—growth, yield, and generative development [8]. High temperature leads to many physiological and biochemical changes. Several effects of higher temperatures have been identified, including reduction of growth, alteration in photosynthesis, inhibition of seed germination, improper development, alteration in secondary metabolism [9,10], and overproduction of reactive oxygen species (ROS) resulting in oxidative stress [8].

*Heracleum sosnowskyi* Manden. (Apiaceae) belongs to the group of “large” or “giant” hogweeds and is listed as an invasive species in the European Union. *H. sosnowskyi* spread rapidly in Poland, the Baltic States, and Russia after its introduction as a fodder plant. It is an undesirable invader due to its large size, high seed production, and vigorous growth, leading to gross changes in vegetation and obstruction of access to riverbanks. The health hazards of this species via serious dermatological effects on skin contact are the main reasons for concern over its spread [11]. Here we studied temperature stress tolerance as a potential contributor to the success of *H. sosnowskyi* and examined the role of SMs and photosynthetic efficiency. In particular, the objectives of this study were to (i) investigate the sensitivity of *H. sosnowskyi* to acute heat stress versus moderate long-term high temperature in relation to oxidative stress markers and photosynthesis efficiency and (ii) evaluate and estimate polyphenolic compounds, especially furanocoumarin levels in the plants under heat stress.

## 2. Results and Discussion

### 2.1. Impact of Heat Stress on Pigments Content and Parameters of Chlorophyll a Fluorescence

Among photosynthetic pigments, chlorophyll *a* (chl *a*), chlorophyll *b* (chl *b*), and carotenoids (car) were measured. To evaluate the chlorophyll *a* fluorescence, the following parameters were established: minimal fluorescence in the dark-adapted state (F0), maximal fluorescence in the dark-adapted state (Fm), maximum quantum yield of PSII in the dark-adapted state (Fv/Fm), photochemical quenching (qP), nonphotochemical quenching of fluorescence (qN), and the ratio of fluorescence decrease (Rfd).

Under the shortest temperature stress (1 d), the decrease in the content of all photosynthetic pigments was significant (Figure 1a). Under longer exposure (≥2 d), both chlorophyll forms and carotenoids had a clear tendency to elevate above the level achieved at the shortest time of stress exposure. In agreement with our results, there is evidence that thermal stress can reduce the total chlorophyll content, e.g., in *Cucumis sativus* [12] or *Artemisia sieberi* alba [13], the chl *a* content in soybean [8], and the chl *b* content in *Valerianella locusta* [14]. Moreover, under high temperature, the denaturation or inhibition of enzymes involved in chl biosynthesis results in the depleted concentration of these pigments [15]. However, the results indicated that increasing pigment content under longer exposure in comparison to the 1 d treatment may contribute to the acclimatization of plants.

Photosynthesis is one of the most heat-sensitive physiological processes [8]. Measurement of chl *a* fluorescence is frequently used in order to assess high-temperature-induced changes in the thylakoid membranes and, as a consequence, PSII damage [16,17,18]. Moreover, among all parameters of chl a fluorescence, Fv/Fm and F0 seem to be good indicators of the function of PSII reaction centers under the temperature exceeding heat tolerance limit [16,18]. Though some studies considered PSII as a strongly temperature-sensitive constituent of the photosynthetic apparatus [19,20] leading to decreased thylakoid membrane stacking, increased fluidity, reorganization of the membranes [18,21,22], and the relocation of PSII light-harvesting complexes from the thylakoid membrane [23], other studies did not provide the same outcomes. PSII cannot be always significantly affected at 35–45 °C [17], which is in agreement with the results of our study. By application of the fluorescence parameters, we proved that plants demonstrated the ability to recover from the temperature stress within a short period (2 d or longer). It can be the evidence for efficiently working machinery for protein biosynthesis in order to protect and repair any thermal damage of PSII and to maintain the balance between oxidants and antioxidants [18,24,25] or resistance against irreversible heat-induced structural changes [17]. Furthermore, the alterations in the levels of lipid desaturation in thylakoid membranes can be associated with the level of thermosensitivity of PSII [26].

The values of two parameters of chlorophyll *a* fluorescence, Fv/Fm and Rfd, are presented in Figure 1b,c and in Appendix A. A temporal decrease in Fv/Fm under short-term stress with subsequent recovery to the control level under long-term exposure was detected (Figure 1b). Initially, a decrease in the Fv/Fm values indicated a reduction in PSII efficiency, mainly by photoinhibition [27], as reported, e.g., in short-term heat-stressed potato [28]. Then, under the prolonged action of adverse temperature, the plant showed strong resistance to photoinhibition, reaching the same level of Fv/Fm under the long-term exposure as the control plants. Similarly, in our previous study [14] with *Valerianella locusta* exposed to long stress, the Fv/Fm ratio was stable and unaffected by the temperature. On the other hand, the Rfd index decreased regardless of the level and duration of the applied temperature stress (Figure 1c). This parameter seems to be more sensitive to the high temperature, which can be explained by the lower rate of photosynthetic CO_2_ assimilation under such conditions [29].

Both F0 and Fm showed an increase only after 1 d of heat stress with recovery to the control level under longer exposure (Appendix A). F0 is a convenient parameter for monitoring the high-temperature impact on PSII inactivation [16]. An increase in the F0 value is observable under various stresses, such as water deficiency [30] or salt stress [31]. In the present study, the reversible increase in F0 (under 35 °C/1 d treatment) could be caused by the temporal inactivation of the PSII reaction center [32] or by the reduction of the primary plastoquinone (PQ) acceptors (Q_A_) through PQ in the dark [16] with the subsequent recovery of the inhibition just after 2 d. The reason for the F0 increase is species dependent [16]. Moreover, the temporal elevation in the F0 value was not associated with the reduction in Fm as it was presented in other studies [33], thus no impairment of the light-harvesting complex of PSII can be suggested. Additionally, an increase in F0 after 1 d of heat stress was connected with the decline in Fv/Fm, which is in accordance with other results [34].

The qN shows the part of absorbed radiation that is not used for electron transport in photosynthesis, and besides the conversion of violaxanthin to zeaxanthin in the xanthophyll cycle, the photosystem II subunit *S*
*protein* (PsbS) protein is also involved in this process [35]. This parameter increased significantly under the 1 d and 2 d stress but decreased under the 7 d exposure (Appendix A). This provides evidence for the acidification of the thylakoid lumen due to the accumulation of protons and, in consequence, to the transitional, short-lasting increase in qN [25]. In agreement with our data, 3 d of heat treatment (but not 6 h of heat) increased qN in potatoes [28]. The qP indicates the proportion of open PSII reaction centers [36]. The value of qP was reduced under shorter heat exposure (1 d and 2 d) and recovered to the control level in the long-term temperature conditions (Appendix A).

The difference in the effects on some fluorescence parameters mentioned in the literature is strongly dependent on the strength and length of the temperature stress chosen for the particular species [16,23], e.g., St. John’s wort [9] or *Cucumis sativus* [12]. We proved that high temperature influencing the content of photosynthetic pigments and chl *a* fluorescence can have an effect on the thylakoid membrane with PSII photochemistry, which is in agreement with other findings [7,23].

### 2.2. High-Temperature-Induced Changes in Lipid Peroxidation and the H_2_O_2_ and Proline Content

Heat stress in plants is a complex function of the magnitude, duration, and rate of temperature rise, which can lead to irreversible damage to plant metabolism [7]. One of the first heat stress symptoms is the overproduction of reactive oxygen species (ROS), which induces the peroxidation of membrane lipids [37]. Here we found that both acute heat stress and longer stress induced H_2_O_2_ accumulation. As shown by the H_2_O_2_ visualization, the accumulation of H_2_O_2_ increased in the order: control < 35 °C/1 d < 35 °C/2 d < 30 °C/7 d (Figure 2). A similar increase in the content of thiobarbituric-acid-reactive substances (TBARS) stimulated by high temperature, i.e., by a ca. 4-, 5-, and 7-fold increase for 35 °C/1 d, 35 °C/2 d, and 30 °C/7 d, respectively, was observed (in comparison with control, Figure 3a).

The accumulation of TBARS is a common indicator of ROS-induced damage to cell membranes as an indicator of oxidative stress processes [14]. High temperature stress has been found to induce membrane lipid peroxidation in many plant species [37,38], and the level of lipid peroxidation is correlated with the duration of heat stress [39]. However, it should be noted that the rate of lipid peroxidation may vary. Jiang and Huang [39] described lipid peroxidation beginning after 18 days of heat stress treatment of two grass species, but cell membrane damage was also observed after one day of heat treatment [40].

The common plant response to heat stress involves an accumulation of osmolytes such as proline [7,15]. The short-term acute stress in both variants caused an approx. 3-fold increase in its content, while an 8.4-fold higher amount of proline compared to the control was found under longer/7 d treatment (Figure 3b). Although some evidence has indicated that the overaccumulation of proline can be toxic to plants [41], this compound can also be responsible for stabilization of the structure of proteins, cell membrane protection, and ROS scavenging [42]. Additionally, proline plays a crucial role in maintaining water status in stressed plants [39]. Our study showed, surprisingly, that both the 30 °C/7 d and 35 °C/2 d stress did not decrease the percentage content of water but even slightly elevated it (Appendix A). Although the present results are in contrast with previous studies [14,15], which showed a decrease in the water content in heat-stressed plants, we can suppose that the high accumulation of osmolytes and the higher flexibility of cell walls under heat stress [43] allowed the accumulation of higher amounts of water by leaves.

### 2.3. Impact of High Temperature on SMs and Antioxidant Capacity

Several studies have documented an increase in SMs or changes in the chemical profile within specimens of the same plant species growing under different biotic or abiotic stress factors [44,45]. There is evidence that high temperature stress exerts great effects on the production and accumulation of polyphenolic compounds [7]. It has been pointed out that the induction of phenolic biosynthesis in plants exposed to heat stress is one of the acclimatation processes [46]. One of the hypotheses has suggested that an increase in SMs in stressed plants is a result of growth inhibition and allocation of fixed carbon into secondary metabolism [46].

*H. sosnovskyi* is well-known for the accumulation of furanocoumarins [47]. Our study showed that both 35 °C/2 d and 30 °C/7 d treatments significantly increased the concentration of xanthotoxin (by ca. 80%) and bergapten by ca. 60% and 120%, respectively (Figure 4). Additionally, isopimpinellin was detected only in the heat-exposed plants (Figure 4b). Previous studies showed a stimulating effect of different abiotic stresses, including UV radiation, drought, heavy metals, salinity, air pollution, and biotic stress (insect damage, fungal or bacterial infection) on furanocoumarin content in different plant species ([48] and the references therein). Bourgaud et al. [49] showed that *Psoralea cinerea* exposed to 32 °C accumulated three-fold and eight-fold higher levels of angelicin and psoralen, respectively, compared with plants cultivated at 21 °C. Similar findings were reported in *Bituminaria bituminosa* investigations [48]. The authors reported that higher temperature increased the content of furanocoumarins in roots and leaves of a Calnegre population of *B. bituminosa* cultivated in hydroponic culture [48]. However, it should be noted that another population, Liano del Beal (population with lower cold sensitivity than Calnegre), did not show changes in furanocoumarin accumulation in response to heat stress [48]. It therefore seems that furanocoumarins are variously regulated by heat stress in the *H. sosnovskyi* leaves, and further field studies are needed. Quantitative changes of furanocoumarins are also affected by other abiotic stressors, e.g., SO_2_ excess depletes their content but with various ontogenetic differences [50]. On the contrary, dramatic stimulation of furanocoumarin accumulation occurs after application of some plant hormones such as methyljasmonate or salicylic acid [51], so the role of these metabolites under heat stress remains to be elucidated as an eventual tool for suppression of these dangerous metabolites in *Heracleum* plants.

The short-term heat stress (35 °C/1 d) doubled the content of kaempferol (Figure 5a). This phenomenon was also observed at the level of soluble flavonol content (Table 1). Elevated flavonol amounts in plants exposed to temperature stress were also observed in *Solanum lycopersicum* [52] or in *Vigna angularis* [53]. In potatoes (*Solanum tuberosum*), both short and prolonged heat stress caused the upregulation of precursors of flavone and flavonol biosynthesis [28]. On the contrary to flavonols, the content of chlorogenic acid (Figure 5b) and soluble phenols decreased in *H. sosnovskyi* under longer treatment (30 °C/7 d), which is in accordance with data from watermelon plants [46]. Both flavonoids and phenolic acids are regarded as efficient antioxidant agents [44]. Here we also found a decrease in antioxidant capacity with the reduction of chlorogenic acid in the leaves exposed to the 30 °C/7 d treatment (Table 1). This finding may suggest that long exposure to high temperature disturbs this nonenzymatic antioxidant system, which was also reflected in the higher oxidative damage described above. In contrast, the 7 d treatment resulted in an almost 2.5-fold increase in the accumulation of anthocyanins: a significant elevation was also observed in the 35 °C/2 d treatment (Figure 5c). Previous studies provided ample evidence for the significant effect of temperature stress on the anthocyanin concentration in plant tissues [7]. However, in contrast to our results, earlier reports indicated that elevated temperature decreased the synthesis and stability of these pigments in buds and fruits [54,55]. Nevertheless, it should be noted that this phenomenon was observed in nonphotosynthetic organs, while vegetative tissues exposed to heat stress demonstrated a tendency towards the accumulation of anthocyanins [7]. Because anthocyanins participate in a reduction in the leaf osmotic potential, which is directly related to the improvement of water status in plants combined with their antioxidant activities [56], the present results indicate that accumulation of anthocyanins is one of the potential heat stress tolerance mechanisms.

### 2.4. Principle Component Analysis

Principal component analysis (PCA) of the selected SMs, antioxidant capacity, proline, and lipid peroxidation level (Figure 6) indicated clearly separated groups according to the experimental treatments used. The first component explained almost 48% of variety total variety, and it was largely determined by all tested variables, except for soluble flavonols and kaempferol. The antioxidant capacity, soluble phenols, and content of chlorogenic acid were positively correlated with PC1, while furanocoumarins, anthocyanins, proline, and lipid peroxidation expressed as thiobarbituric-acid-reactive substances (TBARS) were negatively correlated with PC1. PC1 facilitated the separation of the control from the long-term heat stress individuals, with the latter characterized by a higher accumulation of furanocoumarins, anthocyanins, and proline and a higher level of lipid peroxidation. In turn, the nonstressed plants contained higher levels of antioxidant compounds, including soluble phenols and chlorogenic acid. Plants exposed to both the 1 d and 2 d stress (35 °C) exhibited intermediate values of the analyzed parameters in comparison to control and 7 d stress. However, the plants in the 35 °C/1 d treatment were slightly influenced by the PC2 factor and were positively correlated with soluble flavonols and kaempferol.

## 3. Materials and Methods

### 3.1. Plant Materials and Experimental Design

Mature fruits of *H. sosnowskyi* were sampled in the Lublin city area in September 2019. Seeds were collected from 10 randomly selected plants. The collected seeds showed a similar degree of ripeness, as indicated by their color and hardness. To break seed dormancy, cold stratification pretreatments at 0 °C on glass Petri dishes in moisture vermiculite were applied for ten weeks. After conditioning, five replicates of 15 seeds were sown on the surface of moist vermiculite and incubated simulating a day/night cycle (12 h light/12 h dark) under a temperature regime (22/15 °C) and humidity 60% for 30 days in a growth chamber. When they formed the first leaf, the seedlings were transferred into pots (one plant per pot) filled with garden soil. Cultivation was carried out in a growth chamber at 18/25 °C (8/16 h dark/light photoperiod) under light-emitting diodes at a photosynthetic photon flux density of 150 µmol m^−2^ s^−1^ and relative humidity of 60–65% for 60 days. When 3–4 mature leaves were formed, the plants were divided into three groups (10 plants per treatment): the control (18/25 °C, 7 days), short-term temperature stress in two variants: A (18/25 °C, 6 days followed by 1 day with 28/35 °C) and B (18/25 °C, 5 days followed by 2 days with 28/35 °C), and longer heat stress (23/30 °C, 7 days). Preliminary experiment showed that plants did not grow well at 35 °C/7 d, then temperature 30 °C was tested for longer exposure. During the cultivation, plants were watered to maintain constant soil moisture. For metabolic analyses, leaves from one plant were powdered in liquid nitrogen to make aliquots for subsequent analyses. Four individual plants were analyzed for each parameter.

### 3.2. Photosynthetic Pigment Content and Chlorophyll Fluorescence Measurement

The leaves samples were homogenized in 80% acetone and centrifuged, and the content of chlorophyll a, b, and carotenoids was measured spectrophotometrically (UV-160A Shimadzu, Tokyo, Japan) and calculated according to Wellburn [57].

Chlorophyll a fluorescence measurement belongs to noninvasive and highly species-specific techniques used to assess the photosynthetic performance of plants [36,58]. The parameter was measured using Closed FluorCam FC 800-C (Photon Systems Instruments, Brno, Czech Republic). Exactly before the measurement, parts of the leaf were dark-adapted for 30 min to relax the reaction centers. The following parameters were established: minimal fluorescence in the dark-adapted state (F0), maximal fluorescence in the dark-adapted state (Fm), maximum quantum yield of PSII in the dark-adapted state (Fv/Fm), photochemical quenching of fluorescence (qP), nonphotochemical quenching of fluorescence (qN), and the ratio of fluorescence decrease (Rfd).

### 3.3. Detection of H_2_O_2_ and Determination of Lipid Peroxidation and Free Proline Accumulation

The visualization of hydrogen peroxide (H_2_O_2_) in the *H. sosnovskyi* leaves was performed using 3′,3-diamionobenzidine (DAB) according to the method of [14].

The level of lipid peroxidation was estimated based on the content of thiobarbituric-acid-reactive substances (TBARS) according to the method described previously [59].

The free proline concentration was determined spectrophotometrically as described before [60].

### 3.4. Determination of Secondary Metabolites and Antioxidant Capacity

The *Heracleum* leaf samples were air-dried and ground into fine powder. In total, 0.25 g of air-dried material was extracted three times with 80% methanol (2 × 2 mL) in an ultrasonic bath (Sonorex Digitec DT100/H, Bandelin, Berlin, Germany) for 30 min. The extracts were centrifuged (Eppendorf 5804 R, Hamburg, Germany) at 4500× *g* for 5 min. Prior to the analysis of metabolites, the supernatants were filtered through 0.22 µm nylon filters (Kinesis, Cambridgeshire, UK).

Measurement of the furanocoumarin content was conducted using an Agilent 7100 capillary electrophoresis system equipped with a diode-array detector (Agilent Technologies, Santa Clara, CA, USA) according to the method described previously [61]. Briefly, a fused-silica capillary (50 µm i.d. and 64.5 cm total length) (Agilent Technologies, Santa Clara, CA, USA) was used for the analysis. The separation was performed in 50 mM sodium tetraborate buffer containing 65 mM sodium cholate and 20% methanol.

The content of chlorogenic acid was measured using an RP18e LiChrospher 100 column (Merck, Darmstadt, Germany) (25 cm × 4.0 mm i.d., 5 µm particle size) according to the procedure published previously [62].

Before kaempferol measurements, the methanolic extract was hydrolyzed at 80 °C for 120 min using 6 M HCl. After acid neutralization with a 25% ammonium solution, the mixture was dried and dissolved in water. Prior to HPLC measurements, the solution was purified using solid-phase extraction columns (Lichrolut RP-18, Merck, Darmstadt, Germany). HPLC analysis of kaempferol was performed using a Kinetex C18 reverse-phase column (10 cm × 4.0 mm i.d., 2.6 µm particle size) (Phenomenex, Torrance, CA, USA) according to a method described previously [63]. HPLC measurements of both kaempferol and chlorogenic acid were performed using a VWR Hitachi Chromaster 600 chromatograph equipped with a UV–Vis DAD detector (Merck, Darmstadt, Germany).

The quantification of metabolites was performed from calibration curves prepared with standard compounds (Sigma-Aldrich, St. Louis, MO, USA).

The total concentration of anthocyanins was determined colorimetrically after extraction with acidified methanol (1% HCl, *v*/*v*). The total anthocyanin amount was expressed as absorbance unit corrected by chlorophyll contribution (A_530_—0.25 × A_657_) as described before [64,65].

Soluble flavonol content expressed as mg of quercetin equivalent per gram of dry weight matter was measured based on the formation of a flavonol–aluminum complex [66]. Folin–Ciocalteu reagent was used to analyze soluble phenols, and the contents were expressed as gallic acid equivalents per gram of dry weight matter [67]. Trolox equivalent per gram of dry matter was used to express the antioxidant capacity measured with the use of two types of free radicals: 2-azino-bis-3-ethylbenzthiazoline-6-sulphonic acid (ABTS) and 2,2-diphenyl-1-picrylhydrazyl (DPPH) [68].

In parallel, the residual water content of air-dried samples of *Heracleum* leaves was determined after drying in an oven at 90 °C for 12 h and all data were recalculated per gram of absolute dry weight.

### 3.5. Statistical Analysis

At least 4 individual pots with 1 plant per pot were cultured for each treatment. The experiment was independently repeated 2 times in the same growth conditions. The results were analyzed using a one-way analysis of variance. Before, the normality assumption and homogeneity of variance were checked using Shapiro–Wilk’s test and Levene’s test, respectively. The significant differences between the treatments were estimated with Tukey’s test (*p* < 0.05). The principal component analyses (PCA) were carried out separately for biochemical and photosynthetic parameters (Appendix A). Statistica ver. 13.3 (TIBCO Software Inc. 2017, Palo Alto, CA, USA) was used to perform all statistical analyses.

## 4. Conclusions

The results showed that both short-term and longer heat stress altered the accumulation of SMs in *H. sosnowskyi.* A strong correlation between high temperature stress and furanocoumarin content in the leaves was found. This phenomenon was more noticeable in the 35 °C/2 d and 30 °C/7 d treatments. Interestingly, the accumulation of isopimpinellin was observed only in stressed plants. The study also showed that the accumulation of osmolytes, i.e., proline and anthocyanins, could be one of the mechanisms of heat tolerance in *H. sosnowskyi*. Even short stress (35 °C/1 d) caused a significantly higher decrease in the maximum chlorophyll and photosynthetic yield compared with the other stress treatments. In turn, the lipid peroxidation or H_2_O_2_ accumulation was much more elevated by longer stress as naturally expected.

## Figures and Tables

**Figure 1 ijms-22-04756-f001:**
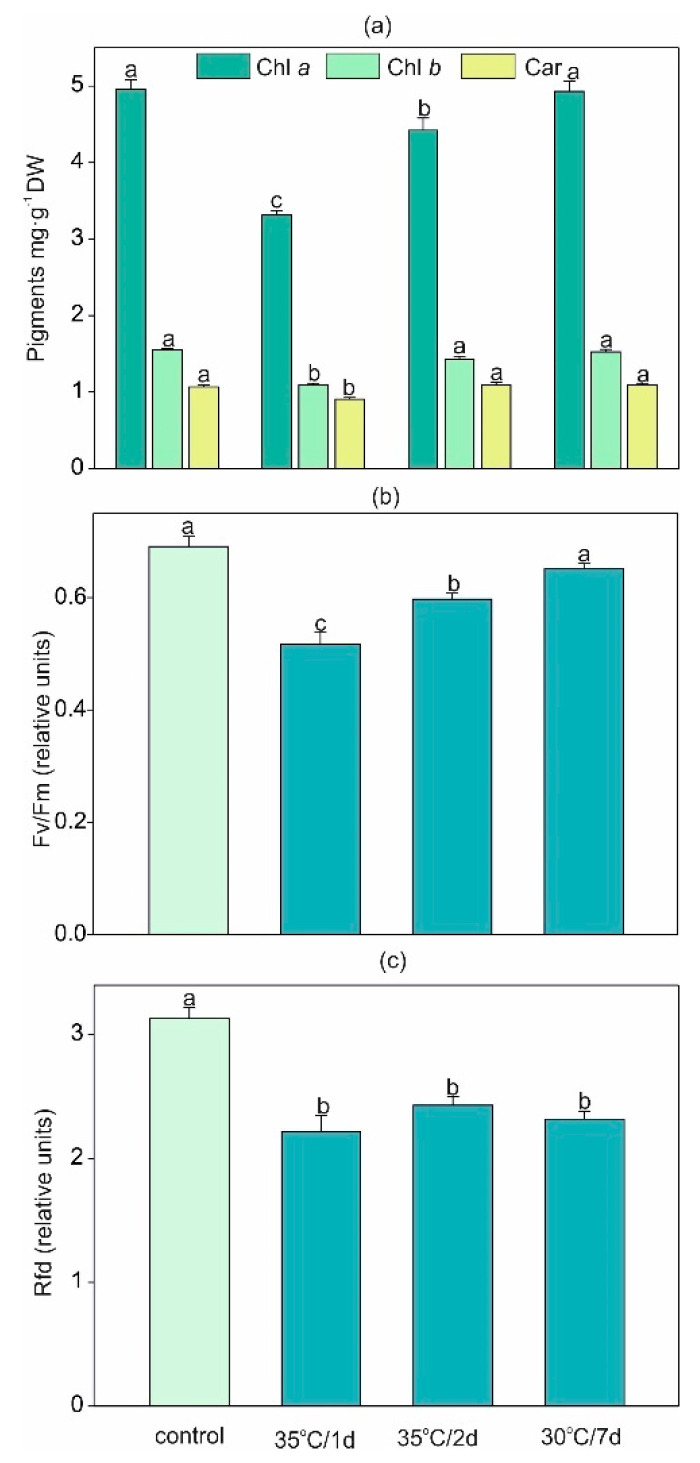
Photosynthetic pigment content: (**a**) chlorophyll *a*, *b* and carotenoids and chlorophyll fluorescence parameters: (**b**) Fv/Fm—maximum quantum yield of PSII/maximal fluorescence in the dark-adapted state, respectively; (**c**) Rfd—ratio of fluorescence decrease in *H. sosnowskyi* exposed to heat stress. Data are means ± SE (*n* = 4 for pigments, *n* = 8 for fluorescence parameters). Values followed by the same letter(s) are not significantly different according to Tukey’s test (*p* < 0.05).

**Figure 2 ijms-22-04756-f002:**
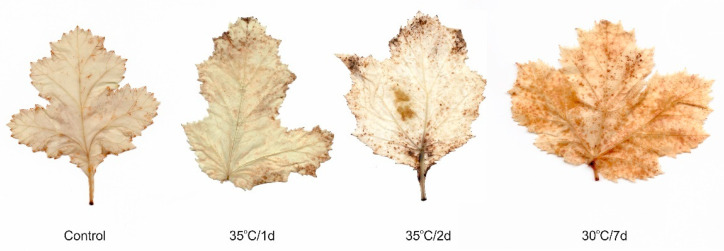
Histochemical detection of H_2_O_2_ using DAB staining in the leaves grown under several thermal conditions. The intense brown color indicates the accumulation of H_2_O_2_.

**Figure 3 ijms-22-04756-f003:**
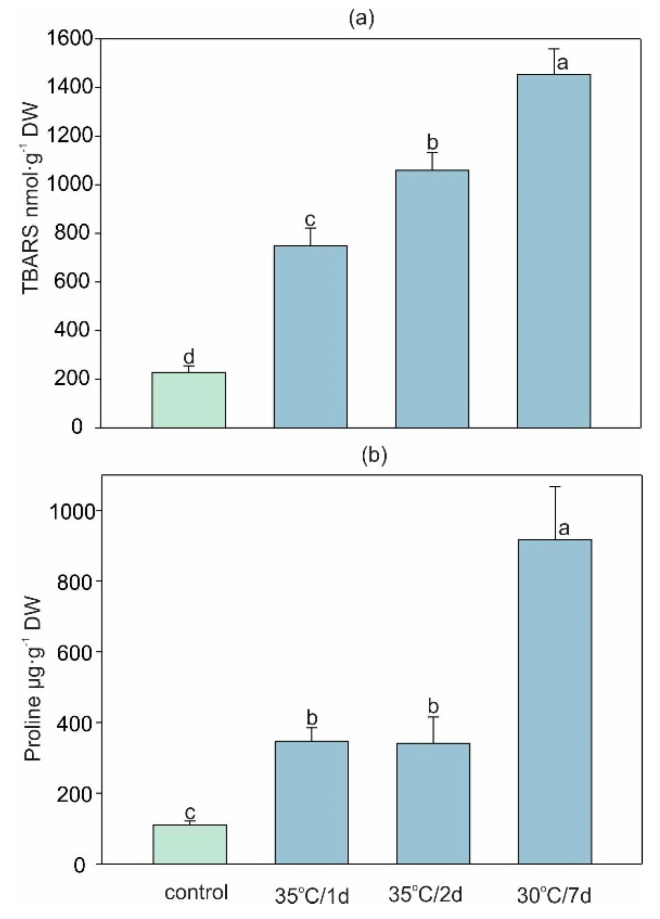
Changes in (**a**) the level of lipid peroxidation expressed by the thiobarbituric-acid-reactive substances (TBARS) content and (**b**) free proline content in leaves of *H. sosnovskyi* in response to heat stress. Data are means ± SE (*n* = 4). Values followed by the same letter(s) are not significantly different according to Tukey’s test (*p* < 0.05).

**Figure 4 ijms-22-04756-f004:**
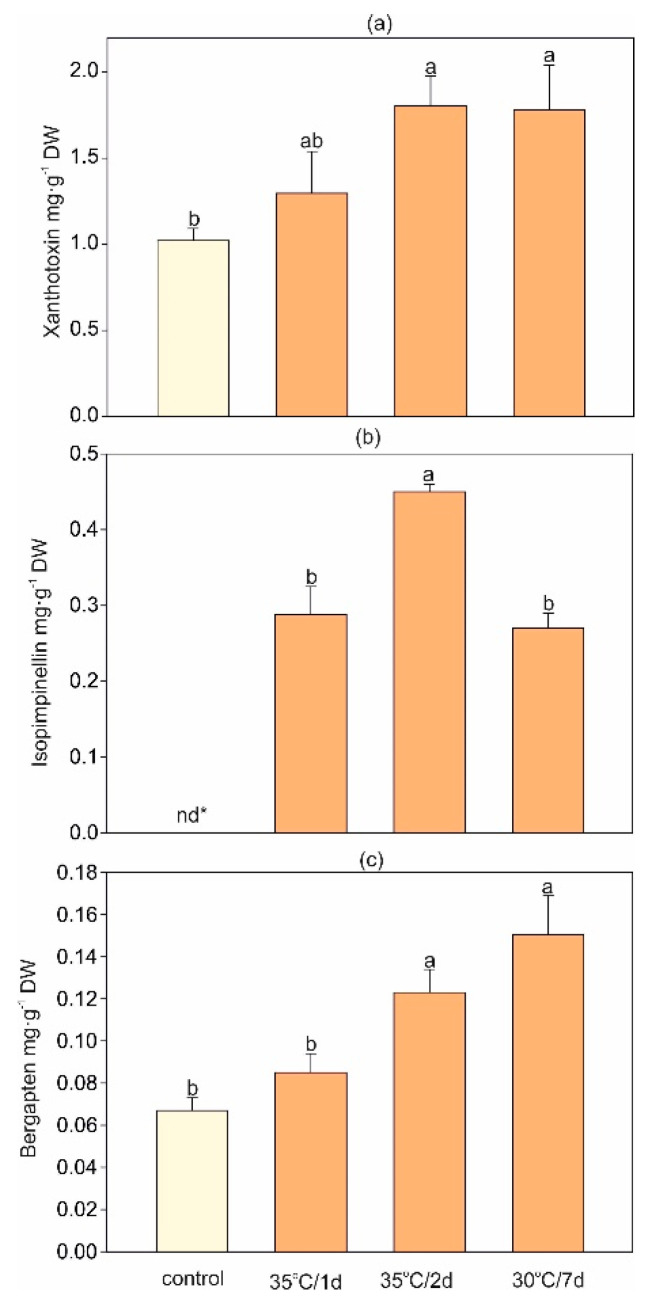
Changes in furanocoumarin contents: (**a**) xanthotoxin, (**b**) isopimpinellin, and (**c**) bergapten in *H. sosnovskyi* in response to stress temperature. Data are means ± SE (*n* = 4), nd*—not detected. Values followed by the same letter(s) are not significantly different according to Tukey’s test (*p* < 0.05).

**Figure 5 ijms-22-04756-f005:**
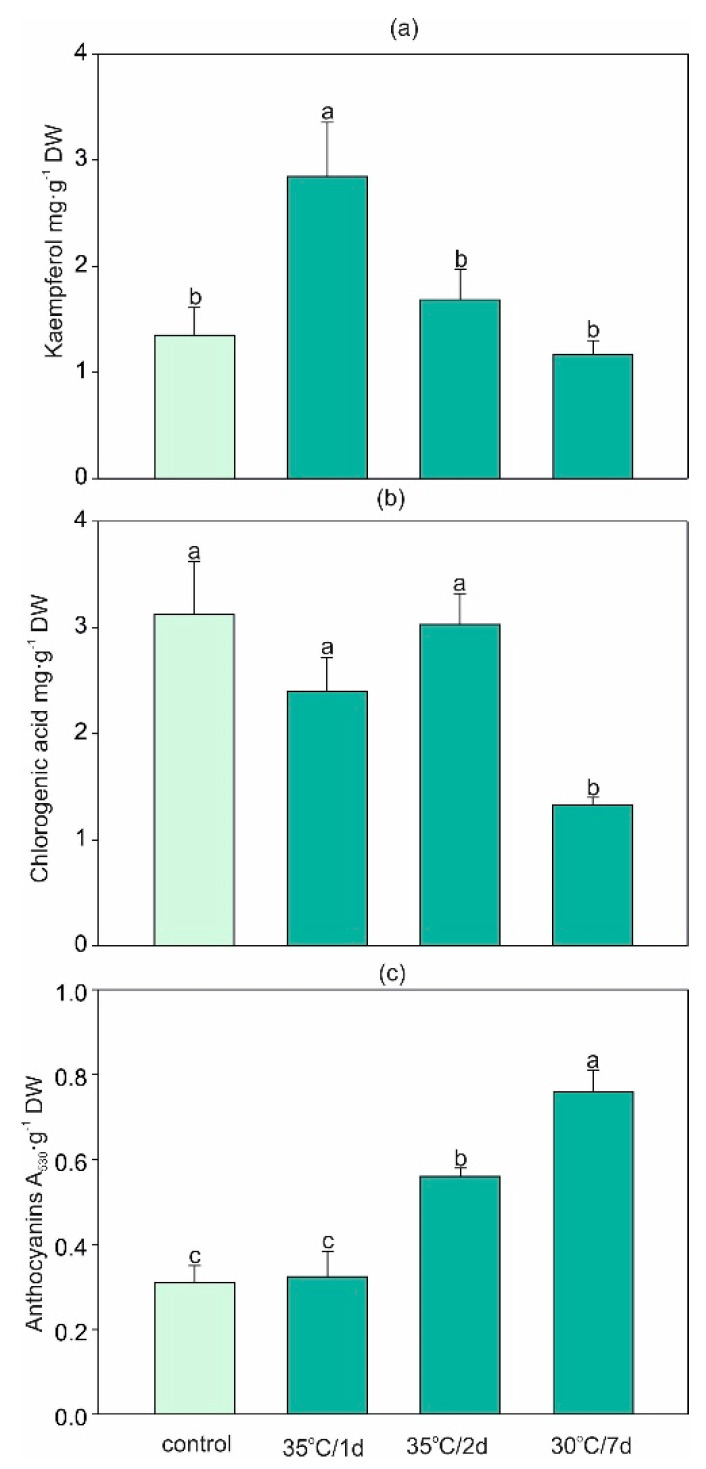
Changes in the contents of selected secondary metabolites (**a**) kaempferol, (**b**) chlorogenic acid, and (**c**) anthocyanins in *H. sosnovskyi* in response to stress temperature. Data are means ± SE (*n* = 4). Values followed by the same letter(s) are not significantly different according to Tukey’s test (*p* < 0.05).

**Figure 6 ijms-22-04756-f006:**
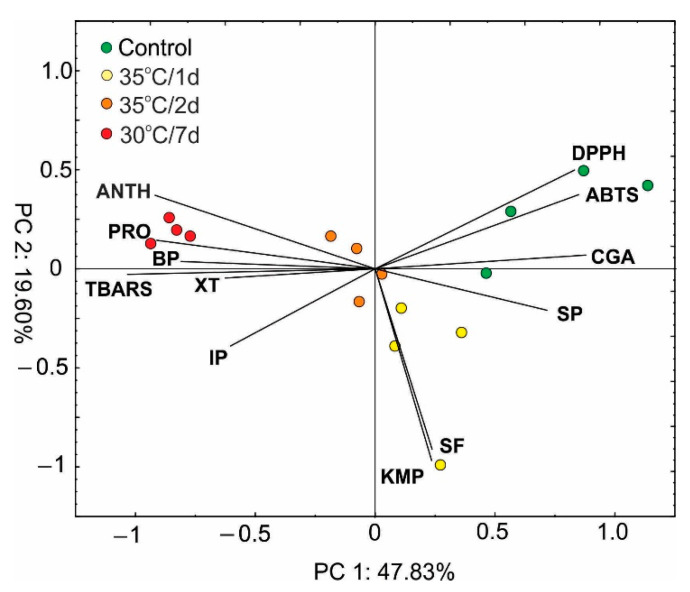
Scaled scatter plot of the principal component analysis of selected secondary metabolites, antioxidant capacity, proline, and the lipid peroxidation level (ANTH—anthocyanins; antioxidant capacity (ABTS, DPPH), BP—bergapten; CGA—chlorogenic acid; IP—isopimpinellin; KMP—kaempferol; PRO—proline; SP—soluble phenols; SF—soluble flavonols; TBARS—thiobarbituric acid reactive substances; XT—xanthotoxin). The length of the lines shows a correlation between original data and the factor axes.

**Table 1 ijms-22-04756-t001:** Effect of high temperature on the content of flavonols and phenols and antioxidant capacity in *H. sosnowskyi*. Data are means ± SE (*n* = 4). Values within the columns followed by the same letter(s) are not significantly different according to Tukey’s test (*p* < 0.05).

	Soluble Flavonols (mg QUE g^−1^ DW)	Soluble Phenols (mg GAE g^−1^ DW)	Antioxidant Capacity (mg TE g^−1^ DW)	
			DPPH	ABTS
Control	0.56 ± 0.04 b	7.97 ± 1.20 a	5.08 ± 0.35 a	10.31 ± 0.61 a
35 °C/1 d	0.85 ± 0.12 a	6.91 ± 0.72 ab	2.96 ± 0.68 b	8.45 ± 0.63 ab
35 °C/2 d	0.62 ± 0.03 b	7.97 ± 0.54 a	3.81 ± 0.26 ab	9.86 ± 0.30 a
30 °C/7 d	0.52 ± 0.07 b	5.27 ± 0.19 b	2.64 ± 0.34 b	7.46 ± 0.30 b

## Data Availability

The data presented in this study are available on request from the corresponding authors.

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
