# Peer review of "High Temperature Alters Secondary Metabolites and Photosynthetic Efficiency in Heracleum sosnowskyi"

_ijms, 2021, doi:10.3390/ijms22094756_

Round 1
Reviewer 1 Report
In line 75, you must write "stress"
If you keep results and discussion together in point 2, then the conclusions correspond to point 4
Author Response
Reviewer 1.
In line 75, you must write "stress"
If you keep results and discussion together in point 2, then the conclusions correspond to point 4
REPLY: We would like to thank the reviewer for a good opinion about our work. Thank you also for all suggestions and corrections. They all were incorporated to the text.
Reviewer 2 Report
The manuscript entitled “High Temperature Alters Secondary Metabolites and Photosynthetic Efficiency in Heracleum sosnowskyi” by Rysiah et al. presents a promising insight about the photosynthetic capacity and secondary metabolism of this invasive species under heat-stress, as an approach to emulate climate change conditions. However, the article has very important flaws that make it not suitable for its publication on IJMS, and I will make a point-by-point justification of my decision:
Abstract
Lines 25, 62: one the worst invasive species. Why? Avoid subjective interpretations and provide only objective information.
Line 27: abbreviations are used in the abstract without expressing their meaning: what does Fv/Fm mean?
Introduction
Lines 69-73: the objectives are clear but objectives number ii and iii are barely redundant, since furanocoumarins are considered as polyphenolic compounds.
Results and discussion
Due to the structure of the journal article, results are provided prior to materials and methods. In this sense, it would be convenient to introduce a brief summary of the methodology performed to obtain the described results.
Figure 1. Authors provide standard error. What’s the point of using the error rather than deviation?
Why authors include the results for photosynthetic parameters as a supplementary figure? How did they bias their work like that? I found these results are as significant as those of chlorophyll content.
What’s the use of determining soluble flavonols and phenols in Table 1 if authors do not use these data for discussion?
Several typing mistakes are found all over the manuscript and, thus, it should be thoroughly revised, i.e.: Line 75: stress*; Line 77: was the significant?; Line 144: “in the leaves of grown in different thermal conditions”?; Line 204: of of.
Material and Methods
The experimental design shows a very important issue: why using a different temperatures between acute and chronic stress? Have authors demonstrated the absence of cut-off effects on plant stress with temperature? There are no explicit reasons for the establishment of this experimental design.
Authors combined different references to perform the extraction and calculate the chlorophyll content. Are they sure that this could be even correct?
To perform the extraction of leaf samples, the authors dried the material at 70 ºC, thus causing the oxidation of polyphenols, driving to a permanent loss of their antioxidant capacity. Such procedures make me think that authors have not made a critical analysis prior to the selection of different methodologies.
Concerning the statistical analysis, authors used 4 pots for each treatment and the experiment was repeated twice. However, almost all results were analyzed with a sample of n = 4 and other with a sample of n = 8. In any case, a sample of four individuals does not allow a proper statistical analysis and the effect of treatments cannot be ensured. How authors determined the individuals to be included in the analysis? Was it a mix of both experiments? Did the authors search for outliers? Did the authors check for normality and/or homogeneity of results prior to the performance of ANOVA? There are a lot of evidences leading to an inconsistent analysis of results.
Conclusions are just a brief summary of the results and lack a useful information since they do not respond to the proposed objectives and do not provide a meaningful significance of the work.
Moreover, there are several concepts that make me wonder about the overall quality of the manuscript. For example, authors use very non-scientific argot, such as:
Lines 25, 62: “H. sosnowskyi is one of the worst invasive species”. In what sense? This is a scientific article, so it should contain only objective and realistic information, and avoid the use of subjective opinions like this one.
Line 82: “the destruction or inhibition of enzymes”. How can an enzyme be destructed? If authors refer to the heat-induced denaturation of enzymes that lead to their loss of function, they should have used more appropriate language.
Line 110, 111: “Both F0 and Fm showed response only after 24 h…”. How can authors express that? Fig. S2 shows that there is a response in all treatments, but such response is significantly higher in the case of 35 ºC/24 h treatment. In fact, any information to discuss F0 and Fm results is provided.
Furthermore, authors thoroughly use several abbreviations without a definition when they firstly appear, as it is the case of Fv/Fm in the abstract and in the result section (lines 27, 93, 94, etc.), chl in lines 81-83, Rfd in line 93, F0, Fm, qN, PsbS, qP, etc. This makes the article very cryptic, and it makes the interpretation very difficult. It seems like the article was rejected in other journal and they only cut and pasted the different parts only for meeting IJMS requirements, without paying attention to the comprehensibility of the article.
Additionally, authors correlate different results in a wrong way, thus driving to incorrect information. For example, in lines 158-166 they declare that “The oxidative stress induced by heat treatments can be related to the disturbance in the photosynthetic electron transfer chain [30], which has been confirmed by the decreased value of Fv/Fm (Fig. 1b)”, but results in Fig. 1b show a totally different pattern from those observed for lipid peroxidation! And the same goes for chlorophyll content and TBARS: “Oxidative stress also affects chlorophyll biosynthesis by inhibiting the activity of enzymes involved in this process [32], consequently affecting the level of this pigment in the leaves (Fig. 1a).” In fact, the treatment showing the highest lipid peroxidation (long-term) does not show any change in the pigments content with respect to control!
In addition, during discussion, authors compare their results with other plant species that are not even related to H. sosnowskyi. Authors should improve the discussion with the comparison to other invasive species if they want to satisfactorily meet the proposed objectives.
I am deeply sorry, but this article needs a deep redefinition, and it should be rejected for publication, since it does not fit the suitable requirements to be classified as a scientific work. In this sense, I hope that authors will take their time to rethink this work and improve the scientific quality of the manuscript and be conscious about the influence they have as members of the scientific community.
Author Response
Reviewer 2.
The manuscript entitled “High Temperature Alters Secondary Metabolites and Photosynthetic Efficiency in Heracleum sosnowskyi” by Rysiak et al. presents a promising insight about the photosynthetic capacity and secondary metabolism of this invasive species under heat-stress, as an approach to emulate climate change conditions. However, the article has very important flaws that make it not suitable for its publication on IJMS, and I will make a point-by-point justification of my decision:
Thank you very much for your time and valuable comments, which all have been considered and incorporated. The detailed list of responses is given below. The changes made in the text of the revised manuscript are marked by red colour. We hope that the modifications and explanation will be acceptable for you.
Abstract
Lines 25, 62: one the worst invasive species. Why? Avoid subjective interpretations and provide only objective information.
REPLY: this is not a subjective opinion but this species is in the list of invasive species of the European Union (e.g. Slovak version of the list: http://www.sopsr.sk/invazne-web/?page_id=821).
Line 27: abbreviations are used in the abstract without expressing their meaning: what does Fv/Fm mean?
REPLY: The abbreviations have been explained
Introduction
Lines 69-73: the objectives are clear but objectives number ii and iii are barely redundant, since furanocoumarins are considered as polyphenolic compounds.
REPLY: I agree, the goals of the work were combined and detailed.
Results and discussion
Due to the structure of the journal article, results are provided prior to materials and methods. In this sense, it would be convenient to introduce a brief summary of the methodology performed to obtain the described results.
REPLY: We agree that it will facilitate the analysis of the presented results. A short list of measured parameters and abbreviations are provided before discussing the results.
Figure 1. Authors provide standard error. What’s the point of using the error rather than deviation?
REPLY: Our study was manipulative experiment with different heat stress treatments used. So we compared average values of different objectives. In our opinion here more properly is showed certainty/uncertainty of the calculated mean values rather than dispersions of a dataset relative to its means. In this cases, the standard error of mean (SEM) or confidence interval (CI) (which directly depends on SEM) are more suitable.
Why authors include the results for photosynthetic parameters as a supplementary figure? How did they bias their work like that? I found these results are as significant as those of chlorophyll content.
REPLY: We have presented 6 different parameters of chlorophyll a fluorescence and divided them into two parts. Two parameters were presented in the main body and another four in Supplementary Material. Transfer of part results into Suppl. Material is normal practice to avoid over-size body of manuscript. As the paper contains numerous data and take into account that the main goal of the paper was to demonstrate the metabolites content we decided to move part of photosynthetic data into supp. mat. like that because our main goal was to demonstrate.
What’s the use of determining soluble flavonols and phenols in Table 1 if authors do not use these data for discussion?
REPLY: The discussion was improved according to suggestion of Reviewer.
Several typing mistakes are found all over the manuscript and, thus, it should be thoroughly revised, i.e.: Line 75: stress*; Line 77: was the significant; Line 144: “in the leaves of grown in different thermal conditions”?; Line 204: of of.
REPLY: Typing errors have been corrected.
Material and Methods
The experimental design shows a very important issue: why using a different temperatures between acute and chronic stress? Have authors demonstrated the absence of cut-off effects on plant stress with temperature? There are no explicit reasons for the establishment of this experimental design.
REPLY: Before the experiment, several thermal conditions were tested. A few groups of plants were tested at uniform temperatures for short and chronic stress. We found that plants did not survive under chronic/7-day stress at temperatures 28/35°C. So, it was decided to reduce the temperature by 5 degrees.
Authors combined different references to perform the extraction and calculate the chlorophyll content. Are they sure that this could be even correct?
REPLY: The literature was checked and improved.
To perform the extraction of leaf samples, the authors dried the material at 70 ºC, thus causing the oxidation of polyphenols, driving to a permanent loss of their antioxidant capacity. Such procedures make me think that authors have not made a critical analysis prior to the selection of different methodologies.
REPLY: This is a misunderstanding that comes from a general description of the methods. The plant material used for the determination of furanocoumarins was air-dried at room temperature. The purpose of drying at high temperature was to obtain data for converting the results to dry matter. Explanations and additions were introduced in the text of the work.
Concerning the statistical analysis, authors used 4 pots for each treatment and the experiment was repeated twice. However, almost all results were analyzed with a sample of n = 4 and other with a sample of n = 8. In any case, a sample of four individuals does not allow a proper statistical analysis and the effect of treatments cannot be ensured. How authors determined the individuals to be included in the analysis? Was it a mix of both experiments? Did the authors search for outliers? Did the authors check for normality and/or homogeneity of results prior to the performance of ANOVA? There are a lot of evidences leading to an inconsistent analysis of results.
REPLY: To obtained appropriate mass of sample, in the case of some metabolic parameters the leaves of the same age were collected from several plants. The leaves were mixed and such prepared combined sample was extracted to analysis of selected metabolites. In order to compare averages between treatments parametric test ANOVA was used. As this test requires normality and homogeneity of variance, the Shapiro-Wilk test for normality and Levine's test for homogeneity were applied.
Conclusions are just a brief summary of the results and lack a useful information since they do not respond to the proposed objectives and do not provide a meaningful significance of the work.
REPLY: just various regulation of individual metabolites we detected is the most valuable outcome of our study which is certainly citable in the scientific community.
Moreover, there are several concepts that make me wonder about the overall quality of the manuscript. For example, authors use very non-scientific argot, such as:
Lines 25, 62: “H. sosnowskyi is one of the worst invasive species”. In what sense? This is a scientific article, so it should contain only objective and realistic information, and avoid the use of subjective opinions like this one.
REPLY: Explanations have been given and the text has been corrected. This species is on official list of invasive species in the European Union.
Line 82: “the destruction or inhibition of enzymes”. How can an enzyme be destructed? If authors refer to the heat-induced denaturation of enzymes that lead to their loss of function, they should have used more appropriate language.
REPLY: The sentence was corrected to be more appropriate.
Line 110, 111: “Both F0 and Fm showed response only after 24 h…”. How can authors express that? Fig. S2 shows that there is a response in all treatments, but such response is significantly higher in the case of 35 ºC/24 h treatment. In fact, any information to discuss F0 and Fm results is provided.
REPLY: To be more specific, we have changed “response” into “increase”. We also added new references into the discussion.
Furthermore, authors thoroughly use several abbreviations without a definition when they firstly appear, as it is the case of Fv/Fm in the abstract and in the result section (lines 27, 93, 94, etc.), chl in lines 81-83, Rfd in line 93, F0, Fm, qN, PsbS, qP, etc. This makes the article very cryptic, and it makes the interpretation very difficult. It seems like the article was rejected in other journal and they only cut and pasted the different parts only for meeting IJMS requirements, without paying attention to the comprehensibility of the article.
REPLY: We have improved this aspect of the manuscript. Now, the abbreviations appear when they are first mentioned.
Additionally, authors correlate different results in a wrong way, thus driving to incorrect information. For example, in lines 158-166 they declare that “The oxidative stress induced by heat treatments can be related to the disturbance in the photosynthetic electron transfer chain [30], which has been confirmed by the decreased value of Fv/Fm (Fig. 1b)”, but results in Fig. 1b show a totally different pattern from those observed for lipid peroxidation! And the same goes for chlorophyll content and TBARS: “Oxidative stress also affects chlorophyll biosynthesis by inhibiting the activity of enzymes involved in this process [32], consequently affecting the level of this pigment in the leaves (Fig. 1a).” In fact, the treatment showing the highest lipid peroxidation (long-term) does not show any change in the pigments content with respect to control!
REPLY: Indeed, this part of the text is not suitable so we decided to remove it.
In addition, during discussion, authors compare their results with other plant species that are not even related to H. sosnowskyi. Authors should improve the discussion with the comparison to other invasive species if they want to satisfactorily meet the proposed objectives.
REPLY: The authors understand the principles of discussion of research results; however, we have a special situation in this case. One of the main objectives of the study was to demonstrate the physiological response reflected in qualitative and quantitative changes in the content of photosynthesis pigments and their fluorescence and the synthesis of secondary metabolites in response to high temperature stress, which can be regarded as additional abilities or adaptations of the invasive species to dispersal.
In the botanical, ecological, and phytosociological literature, the mechanisms of invasiveness are discussed from other perspective, mainly as the presence of certain features of the life history of a species, e.g. the mode of reproduction, seed size and type of seed bank, and morphological traits of vegetative organs, which are analyzed in various groups of plants with such potential.
The experiment presented in the paper intended to show that invasive features also include modifications of secondary metabolism and physiological processes, e.g. photosynthesis. Considering that various data from various species are often compared with model plant Arabidopsis thaliana, it is fully legitimate to compared metabolic responses with other species under identical stress impact (heat stress in this case).
Reviewer 3 Report
The current manuscript does not adequately explain the news in plant regulation mechanisms under high-temperature stress. The quality and novelty of the manuscript are low. Authors need to emphasize the study novelty with in-depth literature review in the introduction. I would have expected a more critical discussion. The Paper is very pragmatic i.e. the decreased Fv/Fm and pigments, accumulation of osmolytes, proline anthocyanins, however, changes in the contents of selected secondary metabolites are new information, but specific for the Heracleum Sosnowski species, not for the understanding of mechanisms (Rubisco, OEC, photosynthetic electron transport parameters...).
Some arguments need clearer and tighter presentation. My major critical remarks refer to the complexity of the paper and to missing information about high-temperature tolerance in photosynthetic mechanisms. The understanding of mechanisms is limited, as it is restricted to papers that have a particular view and deliberately ignore alternatives and results published in reputed journals (S. Allakhverdiev, E. Carmo-Silva, M.Zivcak, C.E.Moore, X.Yang, Y. Song, etc.) and does not present a balanced view of the evidence. The discussion is not satisfactory and needs detailed explanations.
I am sorry but I can not positively support the publication of that manuscript in IJMS.
Author Response
Reviewer 3.
I am deeply sorry, but this article needs a deep redefinition, and it should be rejected for publication, since it does not fit the suitable requirements to be classified as a scientific work. In this sense, I hope that authors will take their time to rethink this work and improve the scientific quality of the manuscript and be conscious about the influence they have as members of the scientific community.
The current manuscript does not adequately explain the news in plant regulation mechanisms under high-temperature stress. The quality and novelty of the manuscript are low. Authors need to emphasize the study novelty with in-depth literature review in the introduction. I would have expected a more critical discussion. The Paper is very pragmatic i.e. the decreased Fv/Fm and pigments, accumulation of osmolytes, proline anthocyanins, however, changes in the contents of selected secondary metabolites are new information, but specific for the Heracleum Sosnowski species, not for the understanding of mechanisms (Rubisco, OEC, photosynthetic electron transport parameters...).
REPLY: we carefully improved the text and added additional references for comparison with other species under heat stress and some parameters are really similar, indicating that our findings have practical impact on the heat stress physiology.
Some arguments need clearer and tighter presentation. My major critical remarks refer to the complexity of the paper and to missing information about high-temperature tolerance in photosynthetic mechanisms. The understanding of mechanisms is limited, as it is restricted to papers that have a particular view and deliberately ignore alternatives and results published in reputed journals (S. Allakhverdiev, E. Carmo-Silva, M.Zivcak, C.E.Moore, X.Yang, Y. Song, etc.) and does not present a balanced view of the evidence. The discussion is not satisfactory and needs detailed explanations.
I am sorry but I can not positively support the publication of that manuscript in IJMS.
REPLY: we regret to read your negative opinion and we did all the best to improve our work. Mainly metabolic analyses provide crucial novelty of our work. Additional remarks of Rev. 3 have also been incorporated and we sincerely hope that explanations and modifications will be acceptable.
Round 2
Reviewer 2 Report
The authors have provided a wider and more convenient background information and it responds to the objectives proposed and they have added the basis for the determination of different photosynthetic parameters.
However, I already expressed my intention to reject this manuscript because their serious flaws. I do not know the causes that this journal made to give it a new review round… I am adhering again to my previous decision since there are still serious problems with the scientific quality of this manuscript.
There are still different abbreviations that have not been defined, such as PsbS (line 138). Furthermore, the use “Acclimation” and “Acclimatation” indifferently and they did not pay attention to use the term to refer to the scientific concept.
Moreover, authors have selected some of the results to appear as supplementary material, just giving a reason to make the manuscript shorter. There is not a specific nor scientific reason to consider some results “important” to be available in the main body of the article and the others “secondary” to appear in the Supplementary Material section. Authors should decide which parameters are important for their objective and not only justifying that they do not add some of the results to the manuscript just to prevent an excessive length of the manuscript.
In addition, there are some sentences that are still vague and subjective since no evidence have been added. For example, in lines 143-149, the authors say: “The qP indicates the proportion of open PSII reaction centers [35]. The value of qP was reduced under shorter heat exposure (1d and 2d) and recovered to the control level in the long-term temperature conditions (Suppl. Fig. 2 d). The change in qP was caused by closure of the reaction centers, resulting from saturation of photosynthesis by light.” How can they affirm that? The result obtained and the justification are not related, and in addition, they have not provided any evidence nor reference for this justification. Then, in lines 147-149, they pass (in the same paragraph), from referring to qN and qP (lines 136-146) to finishing talking about Fv/Fm without relating the different parameters. Details like this make the manuscript less comprehensible, and no clear conclusion is obtained from the determination of photosynthetic parameters, thus losing the importance of this study.
I already hit the attention on the statistical analysis of this work. n values are very low to obtain a robust result. Authors did not pay attention to that and there are still some flaws that should have been reviewed in depth. It should be necessary to include the results for the statistical analyses as supplementary material, since it is very difficult to “believe” that statistical differences are existing, for example in Fig. 3a, where standard error represents more than 10% of the value for xanthotoxin content.
Moreover, regarding the concentration of compounds found, I do not find a reason to use furanocoumarins as markers of plant stress via secondary metabolism induction, since values are less than 2 mg per gram of dry weight, especially in the case of bergapten, where differences less than 0.1 mg have been identified between the different treatments. Which was, then, the resolution and accuracy of the analytic measurements?
Author Response
Response to Reviewer # 2
There are still different abbreviations that have not been defined, such as PsbS (line 138). Furthermore, the use “Acclimation”(line 201) and “Acclimatation” indifferently and they did not pay attention to use the term to refer to the scientific concept
Response: Corrected.
Moreover, authors have selected some of the results to appear as supplementary material, just giving a reason to make the manuscript shorter. There is not a specific nor scientific reason to consider some results “important” to be available in the main body of the article and the others “secondary” to appear in the Supplementary Material section. Authors should decide which parameters are important for their objective and not only justifying that they do not add some of the results to the manuscript just to prevent an excessive length of the manuscript. In addition, there are some sentences that are still vague and subjective since no evidence have been added. For example, in lines 143-149, the authors say: “The qP indicates the proportion of open PSII reaction centers [35]. The value of qP was reduced under shorter heat exposure (1d and 2d) and recovered to the control level in the long-term temperature conditions (Suppl. Fig. 2 d). The change in qP was caused by closure of the reaction centers, resulting from saturation of photosynthesis by light.” How can they affirm that? The result obtained and the justification are not related, and in addition, they have not provided any evidence nor reference for this justification. Then, in lines 147-149, they pass (in the same paragraph), from referring to qN and qP (lines 136-146) to finishing talking about Fv/Fm without relating the different parameters. Details like this make the manuscript less comprehensible, and no clear conclusion is obtained from the determination of photosynthetic parameters, thus losing the importance of this study.
Response: The chlorophyll a fluorescence technique is a widespread method used in stress plants analyses. The technique allows to collect several parameters with various degree of importance so part of parameters is presented in Supp. Material. To keep brevity. Criticized statements were removed.
I already hit the attention on the statistical analysis of this work. n values are very low to obtain a robust result. Authors did not pay attention to that and there are still some flaws that should have been reviewed in depth. It should be necessary to include the results for the statistical analyses as supplementary material, since it is very difficult to “believe” that statistical differences are existing, for example in Fig. 3a, where standard error represents more than 10% of the value for xanthotoxin content.
Response: All results in the paper (including the data of Fig.4 a) were tested using Statistica ver. 13.3 (TIBCO Software Inc. 2017) while the signification of differences between treatments was tested using Tukey’s post-hoc analysis. Although we used Tukey’s test (less sensitive compared to Fisher test) the data supported enough to reject null hypothesis. However, the p value in this case was close (but not above) to limit set 0.05.
Moreover, regarding the concentration of compounds found, I do not find a reason to use furanocoumarins as markers of plant stress via secondary metabolism induction, since values are less than 2 mg per gram of dry weight, especially in the case of bergapten, where differences less than 0.1 mg have been identified between the different treatments. Which was, then, the resolution and accuracy of the analytic measurements?
Response: We do not understand the statement of reviewer that only compounds with higher amount (above 2 mg per g DW) could be consider as stress markers. The analytical details have been published in the paper Dresler et al. 2018: Separation and Determination of Coumarins Including Furanocoumarins Using Micellar Electrokinetic Capillary Chromatography. Talanta 187, 120-124, doi:10.1016/j.talanta.2018.05.024
Reviewer 3 Report
The authors have addressed all queries raised while reviewing the manuscript. The manuscript has been revised thoroughly and may be accepted for publication.
Author Response
Response to Reviewer # 3
The authors are grateful to Reviewer 3 for acceptance of the revised version our work.
Round 3
Reviewer 2 Report
Authors have paid attention to the critical points of this article. I recommend its publication.